# Moving towards culturally competent health systems for migrants? Applying systems thinking in a qualitative study in Malaysia and Thailand

Nicola Suyin Pocock[1,2]*, Zhie Chan[1], Tharani Loganathan[3], Rapeepong Suphanchaimat[4], Hathairat Kosiyaporn[4], Pascale Allotey[1], Wei-Kay Chan[2], David Tan[1]

1 United Nations University International Institute for Global Health, Kuala Lumpur, Malaysia, 2 Department of Global Health and Development, London School of Hygiene and Tropical Medicine, London, United Kingdom, 3 Department of Social and Preventive Medicine, Faculty of Medicine, University of Malaya, Kuala Lumpur, Malaysia, 4 International Health Policy Programme, Ministry of Public Health, Nonthaburi, Thailand

* nicola.pocock@lshtm.ac.uk, nicolapocock@gmail.com

**Data Availability Statement:** All relevant data are within the manuscript and its Supporting Information files.

## Abstract

### Background

Cultural competency describes interventions that aim to improve accessibility and effectiveness of health services for people from ethnic minority backgrounds. Interventions include interpreter services, migrant peer educators and health worker training to provide culturally competent care. Very few studies have focussed on cultural competency for migrant service use in Low- and Middle-Income Countries (LMIC). Migrants and refugees in Thailand and Malaysia report difficulties in accessing health systems and discrimination by service providers. In this paper we describe stakeholder perceptions of migrants' and health workers' language and cultural competency, and how this affects migrant workers' health, especially in Malaysia where an interpreter system has not yet been formalised.

### Method

We conducted in-depth interviews with stakeholders in Malaysia (N = 44) and Thailand (N = 50), alongside policy document review in both countries. Data were analysed thematically. Results informed development of Systems Thinking diagrams hypothesizing potential intervention points to improve cultural competency, namely via addressing language barriers.

### Results

Language ability was a core tenet of cultural competency as described by participants in both countries. Malay was perceived to be an easy language that migrants could learn quickly, with perceived proficiency differing by source country and length of stay in Malaysia. Language barriers were a source of frustration for both migrants and health workers, which compounded communication of complex conditions including mental health as well as obtaining informed consent from migrant patients. Health workers in Malaysia used

**Funding:** We are grateful for funding to conduct this research from the Asia Pacific Observatory on Health Systems and Policies (APO) [grant no. N/A] and the China Medical Board's Equity Initiative [IF055-2018]. APO: http://www.searo.who.int/asia_pacific_observatory/en/ CMB Equity Initiative: http://www.equityinitiative.org/ The funders had no role in study design, data collection and analysis, decision to publish, or preparation of the manuscript.

**Competing interests:** The authors have declared that no competing interests exist.

strategies including google translate and hand gestures to communicate, while migrant patients were encouraged to bring friends to act as informal interpreters during consultations. Current health services are not migrant friendly, which deters use. Concerns around overuse of services by non-citizens among the domestic population may partly explain the lack of policy support for cultural competency in Malaysia. Service provision for migrants in Thailand was more culturally sensitive as formal interpreters, known as Migrant Health Workers (MHW), could be hired in public facilities, as well as Migrant Health Volunteers (MHV) who provide basic health education in communities.

## Conclusion

Perceptions of overuse by migrants in a health system acts as a barrier against system or institutional level improvements for cultural competency, in an already stretched health system. At the micro-level, language interventions with migrant workers appear to be the most feasible leverage point but raises the question of who should bear responsibility for cost and provision—employers, the government, or migrants themselves.

## Background

Globally there are 277 million migrants and 19 million refugees, including 164 million labour migrants of whom around 30% are working in Low- and Middle-Income Countries (LMIC) [1]. LMICs Malaysia and Thailand are major destination countries for low-skilled migrants working in construction, agriculture, manufacturing, services and domestic work. Malaysia hosts an estimated 5.5 million documented and undocumented migrant workers from Indonesia, Bangladesh, Nepal and Myanmar, while Thailand hosts around 3.9 million migrants, mainly from neighbouring countries Cambodia, Laos and Myanmar [2,3]. Both countries host significant refugee populations (179,000 in Malaysia, 103,000 in Thailand) mainly from Myanmar [2,4]. Despite significant in-migration, little is known about how healthcare providers are responding to challenges posed by these changing patient demographics [5].

Cultural competence is a concept that acknowledges the importance of culture as fundamental to effective communication and interaction within a multicultural environment. In healthcare, cultural competency describes interventions that aim to improve accessibility and effectiveness of health services for people from ethnic minority backgrounds [6]. A culturally competent healthcare system is one that recognises the importance of culture and the dynamics among stakeholders that result from cultural differences and adapt services to meet culturally unique needs [7]. Cultural competency improves the ability of health systems and clinicians to deliver appropriate services to diverse populations leading to better health outcomes and reducing disparities [8]. Cultural competency interventions for migrant service use specifically include interpreter services, migrant peer educators or patient navigators, health worker training on providing culturally appropriate care for migrants, and culturally specific education programs with migrant patients.

To date, most research on cultural competency has focussed on interventions in Western high-income settings, with several studies conducted in the US with Hispanic, Indian and African American populations [9]. Interventions have been poorly defined, with a lack of long-term outcomes, lack of standardised assessment tools measuring cultural competency (partly due to lack of consensus on the definition of cultural competency) [9,10]. In Thailand, studies

conducted to date have mostly been observational and not specific to the multicultural aspect of health services for migrant population at the institutional and system level [11–13]. In Malaysia, one paper examines the multicultural counselling experience in the domestic population [14]. Further research is needed to inform the design of culturally competent health systems in the Southeast Asia/LMIC context.

In Malaysia, recent studies using clinic based surveys examine health profiles of migrants, but do not examine health workers' response to migrants, nor do they examine system features that encourage or discourage health seeking among migrants [15,16]. There is a gap in research around implementation and financing of interpreter services for migrants globally, with most research conducted in health settings in high-income countries [17,18]. Very few studies are conducted in LMICs. In Thailand, one study describes the benefits and challenges of implementing a Migrant Health Volunteer (MHV) program in two provinces [19]. In Malaysia, a recent survey with migrant workers found that those who preferred other languages for clinical communication (not English or Malay) were 2.7 times more likely to delay treatment when severely sick, compared to those who preferred to communicate in Malay [20]. In Thailand and Malaysia to date, there are no dedicated studies on how interpreter services are being provided, despite interpretation being a core element of a culturally competent health system [7].

Anecdotally, migrants in both countries have reported discrimination by health providers [21,22]. Refugees also face significant difficulties accessing the health system. In Malaysia for example, refugees report being charged even higher fees than the specified foreigners fees by public providers, linked to provider discrimination [23,24]. It is well known that both undocumented migrants and those with work permits may avoid seeking care for fear of arrest or deportation, or they may internalise exclusionary arguments that they are "undeserving" [25,26], which usually leads to high rates of self-treatment or use of private clinics [27,28]. Notably, discrimination and extortion by other authorities (e.g. police, immigration officials) is another significant source of stress for migrant workers which further discourages health care seeking [29].

Delayed healthcare seeking is associated with financial constraints in Malaysia [22], where migrant workers are required by the Ministry of Health to enrol in a private insurance scheme, the Hospitalization and Surgical Scheme for Foreign Workers (SPIKPA). However, the total coverage amount is low (20,000 Malaysian ringgit/USD 4,741) relative to foreigner fees charged in public hospitals (which saw 100% increases in 2016, linked to MOH budget constraints), and only documented migrants can enrol [30,31]. Among migrant workers using hospitals in Kuala Lumpur, 79% paid out-of-pocket payments for health treatments [32], with 87% of Bangladeshi migrants in another study not receiving any financial support from employers for treatment [33]. In Thailand, migrants (including undocumented migrants) can enrol in public health insurance schemes with comprehensive benefits packages operated by the Ministry of Public Health (MOPH) and the Social Security Scheme for those who work in the formal sector [34,35]. While the Health Insurance Card Scheme (HICS) has improved migrant workers' access to services and reduced out-of-pocket payments (OPPs), outpatient utilization rates have remained low. Migrants primarily used only inpatient services, which meant that there were high self-treatment rates and many delayed seeking care [36,37].

While there is an emerging literature documenting migrants and refugee health needs in both countries [38–41], little is known about provider strategies to accommodate the needs of these groups, and whether and how policies which promote communication and understanding (e.g. interpreter systems) are in place/adhered to. There is an urgent need for wider understanding about how we can improve migrant utilisation of health systems in Asian countries, including supportive system features and feedback loops, among policymakers.

This paper aims to describe stakeholder perceptions of migrants' and health workers' language and cultural competency, with a focus on Malaysia which does not yet have a formalised interpreter system. For the purpose of this paper, migrants include documented and undocumented labour migrants, asylum seekers and refugees. Findings from Malaysia will be complemented by selected data from Thailand, which is in the process of formalising a nation-wide Migrant Health Volunteer (MHV) and Migrant Health Worker (MHW) scheme. A companion paper will evaluate systems factors affecting implementation of this MHV and MHW program in Thailand, from which lessons can be drawn for implementation of interpreter systems in LMICs including potentially in Malaysia. Both papers feature systems thinking diagrams, to scope potential intervention points to improve cultural competency for migrant service use.

## Methods

Qualitative methods were used in an exploratory, iterative design. We conducted informal discussions with stakeholders in Malaysia and Thailand to identify initial barriers and facilitators of a culturally competent health system in both countries. An initial conceptual framework was drafted, which along with previous literature, informed the development of interview topic guides. Subsequent qualitative interviews collected further data on migrant health provision, challenges and facilitators for a migrant friendly health system broadly. Following thematic analysis of interviews and policy documents in each country, the research team discussed emergent findings, which informed development of the final systems thinking diagrams presented in the Results section. These systems thinking diagrams in this paper were developed using data from Malaysia (please see 'Systems thinking diagrams' below).

### Data collection & sampling

We conducted 37 interviews with a total of 44 policy, Civil Society Organizations (CSOs), industry stakeholders and health workers in Malaysia (Table 1) between April 2018 to August 2019. Fifty interviews were conducted in Thailand, in Samut Sakhon and Ranong provinces, with health workers, staff in non-governmental organization (NGO), policy stakeholders, Migrant Health Volunteers and Migrant Health Workers between November 2018 to April 2019. Participants in Malaysia were sampled purposively from an initial sample frame obtained from a previous migrant health stakeholder workshop [42]. Further snowballing from existing

**Table 1. Participant characteristics in Malaysia (N = 44) & Thailand (N = 50).**

| Code* | Participants' Background | Malaysia | Thailand |
|---|---|---|---|
| | | N | N |
| AC | Academia | 3 | - |
| CSO | Civil Society Organisation | 11 | 4 |
| IND | Industry | 5 | - |
| IO | International Organisation | 4 | - |
| HP | Health Professional | 12 | 18 |
| MW | Migrant Worker | 4 | - |
| POL | Policy Stakeholders | 2 | 7 |
| TU | Trade Union | 3 | - |
| MHV | Migrant Health Volunteer | - | 9 |
| MHW | Migrant Health Worker | - | 12 |
| **Total** | | **44** | **50** |

*'M = Malaysia, T = Thailand before codes in results

participants and Linkedin facilitated recruitment of further stakeholders involved in migrant health. Participants in Thailand were recruited purposively from stakeholders involved in migrant interpreting services in two provinces, representing the central and border areas of Thailand, and policy stakeholders in the Ministry of Public Health (MOPH). Selection criteria for Migrant Health Workers and Migrant Health Volunteers included working areas and years of work experience (aiming for a roughly equal sample of those with less than two years' experience, and those with more than two years' experience, based on the average turnover rate of two years). For other key informants, snowball sampling was employed. Informed consent was sought and obtained from participants in both countries to participate in interviews.

Interviews were primarily conducted by a team of medical doctors and academic researchers in both countries. Interviewers could be perceived as trusted authority figures, particularly with migrant workers, MHW and MHV. To lessen potential power imbalances between researchers and participants, the majority of interviews were conducted in locations and at times of the participants' choosing, in a space they were comfortable in. We emphasized that anonymity and confidentiality would be maintained in study reporting. Migrant participants especially were assured that they could refuse to answer questions or to end the interview at any time.

Interviews were transcribed verbatim and analysed in native languages by the multi-lingual research team. Audio files and electronic transcripts were stored on secure servers, and transcripts were stored securely in locked cupboards in the researcher's offices and secured computers. In Malaysia, all except five interviews were conducted in English, while in Thailand, most of the interviews were conducted in Thai. MHVs who were not comfortable with Thai language were assisted by Myanmar interpreters. Following analysis in both countries, selected quotes were translated to English for presentation in this manuscript and accompanying papers. Given the perceived sensitive nature of the research and to encourage participation, participants were not asked for consent to have their transcripts available beyond the immediate research team.

Ethical approval to conduct the study was granted by the Medical Ethics Committee, University Malaya Medical Centre (UM.TNC2/UMREC-238), the Medical Research and Ethics Committee, Ministry of Health, Malaysia (NMRR-18-1309-42043) and the Institute for Human Research Protection, Thailand (IHRP 530/2561).

## Qualitative analysis

Relevant policy documents, including circulars, memos, guidelines and regulations were collected to contextualise interview findings. Thematic analysis of interview transcripts was conducted in Dedoose qualitative analysis software and Microsoft Excel across the research team. Qualitative findings informed the development of systems thinking diagrams, which were conceptualised after discussions among the research team and drawn using VennSim software. Participant diversity gives rise to different perspectives on the same issue, which were reflected in the systems thinking diagrams.

## Systems thinking diagrams

A systems thinking approach enables understanding of inter-relationships, interactions, and various perspectives of a system, including reflecting on the system's boundaries. Systems reflect dynamic, often unpredictable interactions amongst diverse, constantly adapting parts that continually change in relation to each other and the collective environment [43]. These relationships can be represented via causal loop diagrams, which use reinforcing loops (representing feedback loops that accelerate change) and balancing loops (representing feedback

loops that resist change) to generate systems maps. This approach helps planners to understand possible leverage points for policy, feedback loops, and adverse effects of policy change.

In this paper, we present qualitative findings for Malaysia and Thailand as part of a comparative study of two countries at very different stages of migrant-friendly health systems development. Because of this difference, and given the local contextual complexity, we elaborate systems thinking diagrams for Malaysia only in this paper. The Malaysian health system has no formal cultural competency provisions currently, compared to Thailand, where a semi-formalized interpreter system is in place.

The systems diagrams presented in this paper draw on Malaysian data, which were shared with interview participants at a dissemination workshop in Kuala Lumpur in December 2018 [44]. A companion paper submitted elsewhere offers a macro-level health systems perspective on cultural competency, using the Thai case of ongoing formalisation of an existing interpreter and migrant friendly health system [45].

## Results

We identified four major themes affecting micro-level interactions in the health system for migrant service use: Perceptions of language ability, cultural differences and communications skills; Consequences of language barriers and a non-migrant friendly health system; Strategies to overcome language barriers, and; Challenges and barriers to improving cultural competency. Systems thinking diagrams visualizing interactions that surround language barriers are then presented.

### Perceptions of language ability, cultural differences and communication skills

Language ability was a core tenet of cultural competency as described by participants.

**Migrant worker language ability.** Most stakeholders agreed that language barriers were a problem, which meant that migrant workers received sub-optimal health services. Stakeholders acknowledged various language capabilities by migrant and refugee groups and length of stay in Malaysia, with newcomers experiencing greater difficulties compared to those who had resided in Malaysia for longer periods. Familiarity with the health system was attributed to length of stay and existing social networks. With the exception of new arrivals, the Rohingya were perceived to have greater system familiarity and better language ability than other groups such as the Chin, according to an IO participant:

*"They are living in the cluster amongst the Chin population alone. [They are] unable to speak using a local language, never [have] mixed around with the other [communities]. So, even that when they come here, we [have] asked them: 'Do you have any vaccination record?' They probably [would] not understand what vaccination is about. . . [Most Rohingya] have been here for longer than the [Chin]; they are born here, [their] parents [were] probably also born here. . . they have been here for generations! They know the private clinics [and] they know where the public hospitals are." [M-IO-4]*

In Thailand, health staff perceived that migrant workers nowadays had higher competencies in Thai language compared to the past. Migrants who were fluent in Thai appeared to have better chances to secure well-paid jobs, or to become coordinators between Thai healthcare providers and migrant communities. There were various learning opportunities, from formal training of MHW/MHVs by health facilities, to informal education by temples, charitable organizations, or by self-learning:

*"I know that there is a kind of school teaching Thai language in Samut Sakhon [one of the migrant populated provinces in Thailand]. If you do not know the language [Thai] before, you can start here, and you can study and work in the same time"[T-MHV-3]*

Several stakeholders in Malaysia described differences in language ability by migrant worker nationality, with some even asserting that migrant workers had no issues communicating (M-CSO-3). Overall, Malay was considered an "easy" language to pick up in a short period:

*"Some of the foreign workers ah, you must understand–the people who pick up Malay [language] very fast are–the Bangladeshis [and] Nepalese. I mean the Indonesians they [already do] speak Malay." [M-HP-5]*

*"IND 1: Yeah, but you see, in our situation, normally it's between 3 to 4 months they can start speaking [in the] local language.*

*IND 2: Basic lah. I think Malay is pretty easy to pick up.*

*IND 1: EXCEPT [the] MALAYSIANS! Some of the Malaysians don't know how to speak after Merdeka [National Independence] for how many years? But [the] foreigners can do it!"*
*[M-IND-1,2]*

Two participants described migrant workers as having better language skills than Malaysian citizens. In Malaysia's culturally and linguistically diverse domestic population, the expectation from these participants was that all citizens should be able to converse in Malay, especially if foreigners demonstrated this ability. One migrant participant suggested that the onus was on migrant workers to learn Malay:

*"We don't understand Bahasa [Malay language]. This is our problem. Whatever the employer or the doctor says. . . we migrant workers don't understand!" [M-TU-2]*

This participant implies that migrants should learn Malay in order to be understood by employers and doctors. Another doctor similarly inferred that migrants should be socially adaptable, and learn the native language:

*"It's about being able to adapt to where you are [as a migrant]. We can provide assistance but then, the demand for us to adapt to your cultural and your language ability, I guess that's the problem?" [M-HP-10]*

This doctor went on to question whether focussing on migrants who were more adaptable, compared to the minority who didn't learn the language or local customs, would be fruitful:

*"Well, it depends on their social adaptability. . . they can eat the local food, they speak very fluent Malay, but there are some who don't. So, I guess I can't generalise for all migrants as well; but. . . there (are) always those outliers. And so, the question is: 'Do we actually focus on the outliers or those . . . general ones who are able to adapt?' Which are mostly the majority."*
*[M-HP-10]*

Migrant Health Workers who act as interpreters in Thai health facilities, perceived that interpreting services were very important to overcome language barriers. Some migrant

workers did not know the exact meaning of Thai words, which led to miscommunication with Thai health professionals:

> "When patients who could not speak went to see doctors without interpreters, some of them said they understood. In fact, they did not know how to explain the symptoms in Thai words. For example, they said they had fever [in Thai] but they had other symptoms." [T-MHW-1]

**Health workers' language ability and patient communication skills.**   Two participants remarked that health workers had low English proficiency and that migrants were usually asked if they or anyone they knew spoke Malay to communicate [M-IO-2, M-MW-3]. One trade union participant remarked that migrants didn't have confidence in Malaysian doctors, linked to language barriers both ways:

> "Mostly I get very negative comments about the Malaysian doctors. One, they have a very low confidence. And second is the language barriers. Some they cannot explain properly." [M-TU-3]

Language barriers hindered the communication of complex medical terminology by health workers. One participant recalled an example of a migrant worker perceiving that they had a minor condition, compared to a serious procedure they had to undergo:

> "There are certain terminologies that [the] doctor use: [they] are very difficult to understand. . . you don't know what it actually means. Just to give you an example, this guy has got a lump and he has to go through the procedure. So, when the community worker accompanied him to the clinic, the doctor said: 'No, no, no! Don't worry, it's just a small cut and we will just remove it!' So, he came back and thought it was just a small cut . . . When the appointment day come, he's actually admitted in the hospital. . . and. . . this was considered [a] surgery! So, he didn't interpret it as 'surgery', he just told him [that] it's a small cut, you just go taking a pill. . . You cannot take it at face value that the migrant workers understand everything that is explained by the doctor! Sometimes, they don't even understand, that's also good for you to get the second opinion." [M-IO-2]

Migrant workers' basic Malay language skills, combined with doctors sometimes not adequately explaining severity of conditions or procedures, meant that miscommunication occurred. Some participants inferred that doctors didn't have empathy or patient communication skills to fully explain conditions to migrant workers prior to administering medication or treatment:

> "I think the doctor also don't understand our 'bahasa'(language). . . When we are sick the doctor does not explain what's wrong with us. We don't know what doctor is thinking. . . But he still gives medicine. . ." [M-TU-2] translated from Malay

It was difficult for health workers to communicate empathy with the language barrier according to this participant:

> "In [the] government hospital, well, I think treatment wise, it's not a problem, but you see–caring for a patient is just not [about] giving the pills. It's also talking [to the patient], empathising and advising. How do you give these IF there is a language barrier? You can't do that. So,

*a nurse may come and say, 'okay, makan ini ya! [okay, eat this yeah].' That's all! Because she cannot do anything else! Otherwise, if it's a local patient whom she can communicate [with], [the communication will be better e.g.,] 'mak cik, sudah masa ah, boleh makan ubat ini'. [Mam, it's time to take your medicine.]" [M-HP-6]*

It was easier to show good bedside manner and communication with local patients who spoke Malay. Some participants described doctors looking down on migrant workers because of their poor language abilities, and their impatience in treating migrants as a result:

*". . . because of the communication, and doctors maybe look down on the migrant workers, so they just give the Panadol lah! First of all, you cannot tell your problem in the language that the doctor understands. And then, the doctor doesn't have the patience or [he] has low degree of humanitarian response. He won't take it very seriously. . ." [M-CSO-1]*

Migrants' conditions may not be taken seriously, and prescription of unsuitable medication like Panadol was frequently described among participants. Medical errors were also described, although it is unclear whether these were always attributed to language barriers:

*"I went to the clinic the other day because I had [a] fever. I told the doctor, 'I had fever'. The doctor gave me an injection, after that I could not walk for 5 hours! After that, I went back to the clinic and confronted the doctor. He apologised and admitted he gave me the wrong medication!" [M-TU-2] translated from Malay*

**Employer perceptions of cultural differences with migrant workers.** An industry participant described different familiarity with modern medicine among migrant workers. Indian and the Nepalese workers brought their own medicines into Malaysia, as they had more exposure to modern treatments, while Indonesian workers were described as being more superstitious and reliant on traditional medicines (M-IND-1). Indian workers were not used to seeing doctors at home, instead relying on pharmacies where high dosage medicines were prescribed. It took a while for them to adjust to lower dosages administered by Malaysian health workers:

*"But–that's why–you see, certain workers–Indian workers: for the first year, normally–whatever treatment that we [have] done here–it's not that efficient to them because they are used to taking high doses. . . That's why, normally, Indian workers, after [the] second year they get used to the way Malaysians [are] being treated [medically]." [M-IND-1]*

Indian workers often had these medicines confiscated at customs upon arrival in Malaysia. Elsewhere, there might be high-handed ways of dealing with migrant workers by employers, which arose from the perhaps positive intentions to solve problems, according to a CSO participant. This participant described how employers responded with knee-jerk reactions to migrant workers behaviour that they didn't understand, such as curfews in response to alcohol or drug addiction:

*"The solutions have not always [been] ethical, [employers] really think: 'How do I handle this drug and alcohol abuse problem? Oh [let's do] curfew!' Because [employers] don't have other options. . . the kind of issues these people face on a daily basis: there's no standard SOP for that. How do you handle one worker trying to run off with the other workers' wife? How do you handle someone abusing drugs. . . if you lose him . . . [those] men; your production is*

*completely gone.! How do you handle it? . . . There's all these kinds of issues that these people have to deal with on a daily basis and mental health issues [as well]. . . [Employers would] just receive a group of workers [that] they know literally nothing about beyond their basic bio-data; and then they live in community, which is also the difference with palm [sector]. So, it's very complicated."* [M-CSO-5]

This participant went on to infer that beyond health workers, it was important for human resource staff and middle managers overseeing migrants they hired, to understand cultural differences as well:

*"Sometimes there's no easy solutions for company staff; [as well as] trying to deal with a very diverse group of people [of] different nationalities. Different cultural clashes, [Or they have different] education levels–it's not easy! . . . I think we have to keep that [question] in mind:. . . People that are dealing with migrant workers, even at the company level–not even at the hospitals and medical clinic level, even just HR person, the middle manager: 'Do they know how to really deal with these cultural issues too?'"* [M-CSO-5]

A policy stakeholder similarly felt that cultural differences with migrant workers were a major barrier to understanding with employers and locals more widely. This stakeholder felt that cultural education for migrant workers, before they came to Malaysia, could potentially remedy misunderstandings. This stakeholder compared migrants coming into Malaysia before as being more trained in cultural know-how, to now whereby more incoming migrants and lack of cultural understanding was a concern:

*"From the country of origin, they need to be introduced to cultural knowledge and how to adapt. It has worked well in the past. But somehow when the [high] volume [of migrant workers] gets in, you don't really see that now! I remember those days, Bangladeshis [would] come, they are very polite. Those who have high qualification, the character is good, they plan well, they learn Malay very fast [and] they work with locals–[the] Indonesians as well.*

*But [when] so many come, the volume is there; then you see a bit of differences [in these qualities]. It is hard, if we don't control [immigration], we might have the same feeling of [the] British [people] when they talk about Brexit. That might happen! The sentiment! IF you know, they came in, filling [jobs at] the workplace. Somehow, they are not culturally accepted . . . So, it will create more hate and also AFFECTS the society."* [M-POL-2]

This participant implied that migrants being absorbed into the labour market might generate anti-immigration sentiment among citizens. They went on to describe how Malaysians were accepting of migrant workers, but that cultural misunderstandings posed a threat to societal harmony:

*"We are very accepting [towards] [migrant workers]. Actually, on the cross-cultural issue. . .. as a Malaysian we are already dynamic. We have high tolerance on other races, we don't talk about racism or what not. We are Malaysians but somehow when they [migrant workers] come in, we have another set of cross-cultural issues. They have to respect this society of Malaysia rather than looking at us as different races. . . This cultural issue [is] very, very serious to me."* [M-POL-2]

The implication was that migrants should respect Malaysian culture and adapt accordingly.

## Consequences of a language barriers and a non-migrant friendly health system

**Informed consent and medical errors.**   Several participants noted that language barriers delayed healthcare seeking among migrants, who might present at clinics with late stage serious conditions. Worryingly, language barriers could lead to a lack of informed consent with migrant patients even for serious procedures when they did seek care, as described by this participant:

> *"My friend from Ipoh worked in a plastic factory. He had an accident, and cut his finger. He told his employer: 'I don't want to amputate my finger!' The doctor did not understand [or] maybe the employer told him differently. The worker could not understand the Malay language. So, the doctor amputated! [Below elbow amputation]." [M-TU-2] translated from Malay*

This participant went on to question why below elbow amputation procedure was done when the patient refused and when the injury was on the finger, indicating poor communication from the health provider. As a result of medical errors, perceptions that doctors didn't take their conditions seriously, and sometimes lack of informed consent, migrant workers might develop fear and mistrust of health workers. A public-sector doctor suggested that the threat of legal action was an incentive to ensure patient consent was taken (M-HP-4). Among doctors overall, adhering to professional standards of care was perceived to be important. Migrant CSOs or TU's disagreed in some cases.

**Mental health assessments.**   Language barriers were amplified when doctors had to make more nuanced assessments of a migrant patient's mental health condition. While the MOH Malaysia had developed a detailed mental health screening tool, this was considered impractical to administer with migrant patients because of the language barrier (M-HP-5). Screening for psychiatric illnesses is conducted as part of the mandatory medical screening process for incoming documented migrant workers, which was implemented by FOMEMA, a company appointed by the government to conduct foreign worker's medical screening. In Malaysia, having a psychiatric illness is a deportable condition. But, it was unclear how FOMEMA panel doctors screen for mental health disorders. This doctor went on to describe a simplified, visual form of mental health assessment for migrant workers used for this purpose:

> *"HP-6: So, they [doctors]'simplify' it to get a basic mental assessment.*
>
> *HP-7: Well. . . things are quite good actually. So far, it's okay. . . It's compulsory. So, everybody's doing it. So, you simply–basically are looking at the behaviour, his dressing, the way he talks.*
>
> *HP-6: The manner.*
>
> *HP-7: You know the manner, these kinds of things, you know; [but] no detail [or] that history on his friends and families." [M-HP-6,7]*

Some doctors perceived that diagnosis of mental health conditions via a visual inspection of a patient's manner and condition was accurate enough, rather than administering a validated screening tool in the appropriate language. One doctor went on to explain that panel doctors' remuneration by FOMEMA for conducting migrant worker screenings were very low and had not been revised for two decades, which incentivised doctors against conducting a lengthy, time consuming mental health screening:

*"Secondly, this FOMEMA. . . while we are agreeable to do anything the government wants We are also–not been treated [justly]. A 60 Ringgit [service fee] has been given for the last 20 years*! *So, if you give me a 60 Ringgit [fee], I mean let's be honest. I will do a good job but if you asked me to spend 40 minutes on a mental [health] test, I won't do [it]*! *[M-HP-5]*

According to some doctors, it was easy to infer that migrant workers had mental health conditions, when they often arrived in Malaysia with debt or had made sacrifices to migrate:

*". . .He will say, 'I sold my cow when I came [to Malaysia]; I sold my buffalo and came [to Malaysia].' So, all these are natural [mental health] problems."* [M-HP-5]

It was considered unfair that doctors had to make lengthy assessments without adequate compensation from FOMEMA. In these cases, using an interpreter to infer whether migrants were symptomatic for mental health disorders, would presumably take more time than the visual assessment.

Migrants were concerned about privacy and possibly stigma when disclosing mental health issues, which meant that those attending a mental health clinic wanted interpreters from outside of their home community. As one service provider explained:

*"[That] is such an important thing (because) 'word of mouth' [and] the gossip, which passes between [the community]. . . is something which is very toxic and very hard for them to get through. For example, if we have a Pakistani Ahmadiyya patient that comes in, there are cases where they have said*: *'We would prefer if the person who was translating or the person who was providing us with the service IS NOT from the Ahmadiyya community*! *But it could be someone who is a Pakistani Christian'. . . it's okay to talk then."* [M-CSO-10]

Beyond gender, religious and ethnic sensitivities were considered when choosing interpreters. This participant went on to describe difficulties where equivalent terms did not exist in the migrant's native language for mental health conditions:

*". . .It's also very difficult [to get the terminologies right]. For example, how do you translate 'schizophrenia' into the Burmese language*? *Or even to a Rohingya person*? *That word doesn't exist [in their dictionary]*! *So, even [the term] 'stress' or 'anxiety'; it's also about [the interpreter] having to understand the meaning of it*: *the signs and symptoms. . . behind it [and] to be able to translate it to someone else; and not just interpret it as [it is]."* [M-CSO-10]

This provider had created a glossary of mental health terms for various migrant languages, and system of community health workers to mitigate some of the difficulties in conducting mental health assessments with migrants.

## Strategies to overcome language barriers

**Health worker strategies.**  Doctors had several ways of mitigating language barriers with migrant workers, ranging from use of Google translate to sign language or gestures to try and bridge the language gap:

*"Of course it's difficult if they don't bring someone to help communicate. But then we have Google translate. We just use Google translate and it's somehow working."* [M-HP-4] *translated partially from Malay*

Just one doctor in Malaysia mentioned learning migrant languages in order to communicate (MD-1). While in Thailand, short courses for health workers were provided by Provincial Health Office, MOPH, teaching basic communication in Burmese related to health issues and cultural differences. However, there were concerns about time constraints to attend courses. Burmese accents were difficult for doctors to pick up because of different accents among ethnic groups in Myanmar:

*"Provincial Health Office supported Burmese training course in the weekend for health personnel who were interested...To take patient history, I tried to read Burmese words, but patients [are] still confused because of incorrect accents"* [T-HP-18]

In Malaysia, Doctors usually encouraged migrant patients to bring an English or Malay speaking colleague, or Malaysian partners in the case of some migrant women, along to help interpret during consultations (M-HP-8, M-HP-1). However, in public hospital outpatient departments, time constraints meant that doctors would rather resort to hand gestures rather than request friends to act as informal interpreters:

*"Unless it's in a government hospital, [there's a] line up to 100 patients, you know, I don't have time to call 3 fellas [fellows] to come and do your interview! I already understand you, from what you are telling me, you know, maybe we can even say, you know. Yeah, [it's] cough [or] running nose."* [M-HP-5] [Participant demonstrates hand gestures]

Doctors in inpatient and ICU settings were more likely to search for colleagues or friends to interpret for a patient's history, and resort to treating symptomatically until then. Among informants in Thailand, there was much less confidence expressed in health workers' ability to overcome language barriers without interpreters, via gestures or otherwise. If there were no interpreters in health facilities at that time, Thai health workers similarly resorted to asking patients to find someone who could help interpret nearby:

*"[When patients came to health centre without interpreters], I would ask them to find people nearby who could speak Thai. I would not prescribe drugs [without receiving information] because it was harmful to patients. Sometimes, we knew that they had stomach ache and we could guess [diagnosis] from surrounding contexts. However, we would like to know the exact diagnosis so we usually asked for migrant workers who could speak Thai to interpret coming with patients.'* [T-HP-8]

**Employers strategies.**    Migrant workers were deliberately not given hazardous jobs in plantations due to language barriers inhibiting their understanding of occupational risks, according to this industry participant:

*"[Occupational Safety and Health (OSH) training material is given in] pictorial form and also during the [occupational safety and health] training we (will do) demonstration [on] what they can or cannot do. The bottom line is: usually for–hazardous jobs... we (won't put them there)–mainly our foreigners/[workers] are focusing on ... harvesting and some other general work as well."* [M-IND-2]

Employers also described peer liaisons with Malay language ability on worksites, alongside the Estate Hospital Assistants (EHA) which are mandated by Malaysian legislation to be present on each plantation:

*"So, in terms of language, of course, they have to use some simple language, and they don't get [interpreters]–because there are three or four different type of languages; you can't bring an interpreter for each and (all) but they have [a] person who comes and explains to them: by showing equipment and signs and all that. But they do have regular [training] and then–you have this Estate Hospital Assistant [EHA] in this big plantation, who [would] also give training to their workers regularly; on fire safety [and] on chemical exposure: 'What happens if you have a chemical splashed on your face? What do you do?'" [M-HP-7]*

The Workers' Minimum Standards of Housing and Amenities Act 1990 (Act 446) states that plantation owners have the duty to provide workers and their dependents with medical attendance, care and treatment at the estate hospital, group estate hospital or estate clinic. EHAs were an essential link between migrant plantation workers and service access, but it is unclear if any training was offered to EHAs on the cultural differences between migrant workers on care-seeking or factors that would affect health service use.

Quality of training delivered in native languages on plantations was variable/unknown according to a CSO participant:

*"Sometimes for the trainings, what they do is [that] they will do it by batch of workers. So, they will do like the Nepali batch first. Then worker who's been there longer would be the interpreter for the others. That is quite common that they that they ensure that people understand that way. But what is the quality of the interpretation given by the head worker who has been there the longest? I don't know."[M-CSO-5]*

This participant went on to describe cultural differences in perceptions of health and safety among migrants, which put them more at risk of accidents:

*"We have to all go forward together and there needs to be greater awareness about health and safety . . . It's like . . . we have someone who came to try to fix some lights here in our office. The guy doesn't even turn the light off before starting–he is getting a zap! [But he was like]: 'Ah! Doesn't matter!' It's this mentality." [M-CSO-5]*

Companies did not push OSH training or a strong safety culture as much as they should because workers became aggressive or demotivated in response, according to this participant. A policy stakeholder described how the government OSH agency supported training, recruiting local ethnic Chinese and Indian Malaysians as interpreters, who may then become OSH trainers for migrants from those countries:

*"From our side, we get the trainer who can speak their language. Like currently there (are) a lot of Chinese [migrants] coming in; so, we [do] use local Chinese [interpreter] to match that [demand]. And we introduced the local Chinese to them, then if it works well, then we can use him as a trainer–and he is a registered trainer as well. (whereas for) Indians, we use [the] local Indians [as translators], test it out whether the dialects fit [the audience]–then we [will] carry on. Otherwise we can allow them [clients with Indian foreign nationals] to bring interpreter. Then we [will] check continuously–check on the delivery [standards]." [M-POL-2]*

This participant went on to suggest that Embassies of sending countries like Myanmar should be consulted, to locate interpreters for OSH training.

## Challenges and barriers to improving cultural competency

**Informal interpreter systems at CSO clinics.**   Informal interpreter systems consisted of community members who may be refugees, asylum-seekers or migrant workers informally acting as interpreters in CSO clinics. One participant described improvements that needed to be made, broadly around the professionalisation of informal interpreters. Interpreters needed training on ethics and patient consent:

> *"We do [have] code of conducts, we do [have] ethics, and we talk about all confidentiality, non-disclosure, consent–[which] is a huge thing for us. So, we need to make sure everything is [clear and that] they understand why we need to get consent; how to share information; what [kind of] information can be shared. Even amongst the interpreters. So, for example, let's say the patient disclosed this thing to the doctor in this room: 'You shouldn't share the information with the other interpreter in the next room when you go out for lunch!'"[M-IO-4]*

This participant went on to discuss the challenge of interpreters getting used to regular working hours and the general need for training on professionalism and work culture:

> *"This is something new for them as well. They have not been in [an environment and] working formally. So, this is something that we need 'training' [on]; we need to also get them used to it: 'that you need to "clock-in" in this hour.'" [M-IO-4]*

Furthermore, interpreters had to be familiar and keep up to date with medical terminology and conditions. In CSO clinics with informal interpreter systems, continuing training consisted of talks by specialists.

In Thailand, formal interpreters or MHWs usually worked in health facilities and received formal training. Informal interpreters or MHVs worked in communities. MHVs also received training by healthcare staff, but this training was focussed on basic health education with less emphasis on interpreting skills or healthcare and professional competencies. However, some migrant interpreters expressed interest in learning skills beyond their interpreter role, such as preventive care, basic life support and first aid:

> *"Sometimes they [healthcare staff] performed CPR, I would like to learn and know about it. In the emergency situation, it was unpredictable what we would confront, so I would like to help others." [T-MHW-7]*

Although, some MHWs were eager to acquire more clinical knowledge and work further beyond their interpreter role, some health workers opined that MHWs' roles should be limited only to interpreting functions:

> *"The point is that you cannot advise [patients] because that is the role of doctors. You should not tell patients that they had hypertension 100%, so you could do only measuring blood pressure and interpretation." [T-HP-3]*

In Malaysia, two main problems were identified with hiring asylum seekers or refugees as informal interpreters: the inability of CSOs or IOs to formally hire them given Malaysia's legislation prohibiting employment among asylum seekers and refugees, and; what was perceived to be conflict of interest among IOs or UN agencies, whereby informal interpreters were also beneficiaries of their organisations (M-IO-4). Difficulties hiring interpreters present another barrier to improving cultural competency. While formal interpreters can be officially hired by

the MOPH in Thailand, there is no training for informal interpreters or other supporting systems for professionalism.

**"Opening the floodgates" and domestic priorities.** However, participants touched on real and perceived resource constraints to improving cultural competency. One participant alluded to the perception that system improvements for migrants in the form of interpreters would encourage further use of an already stretched health system:

> *"You know, whether they [interpreters] get payment or whatnot; and of course, the other people [will say]: 'Having this, is just–just not only going to attract them? Oh, now not only they can tap into the HEALTHCARE SYSTEM! We allow them [to have access to] TRANSLATOR! They are (going to get better [services].' You know, 'What are you giving them now?'"* [M-POL-1]

This participant described interpreters as a gateway to further concessions or systems improvements for migrants, which the general population would disagree with. One service provider described how domestic healthcare challenges meant that policies on refugees or migrants had less priority:

> *"At the current moment, there is not a huge area of discussion for policy [on] refugees and asylum seekers because we can't even work on our own health issues [in the country]. It's a bit difficult to bridge the topic itself, but we [are] (trying) to get involved in a larger scale with our partners/working groups."* [M-CSO-10]

**Availability of guidelines on cultural competency for health workers.** In Malaysia, participants did not have MOH guidelines on cultural competency to refer to. In contrast, Thailand's MOPH has a medical terminology guide which is translated to the main migrant languages [46]:

> *"It was a book like this [bilingua dictionary]. Sometimes it printed in book while sometimes it was in one-page paper. [When I would like to use it], I opened it to see common words like stomachache, vomit or fever. The common used words were provided in this book."* [T-HP-18]

Generally, doctor-patient manner and cultural competency was perceived to be something doctors learnt on the job, when it was not formally included in medical school curriculums:

> *"During my time I don't think they have this kind of [cultural competency] programmes. I don't think we were taught that much in medical ethics as well–medical law or about all the social behavioural sciences. We were very focused on physical health, not even mental health– I guess, in Malaysia, it's a big taboo [to talk about] mental health. We haven't gone past that yet as well. That's also a big hurdle for us but in terms of this soft skills and cultural competency, I guess it comes with experience."* [M-HP-10]

Cultural competency was also needed for mental health according to this participant, which remained a taboo subject in Malaysia. Learning on the job and coping with existing language barriers and cultural differences with migrant workers was commonly reported by health workers in this study.

## Visualising interactions that surround the language barrier system

Based on thematic analysis findings primarily from Malaysia, Fig 1. shows migrant worker and doctor pathways to addressing language barriers, which inhibit cultural competency in a health

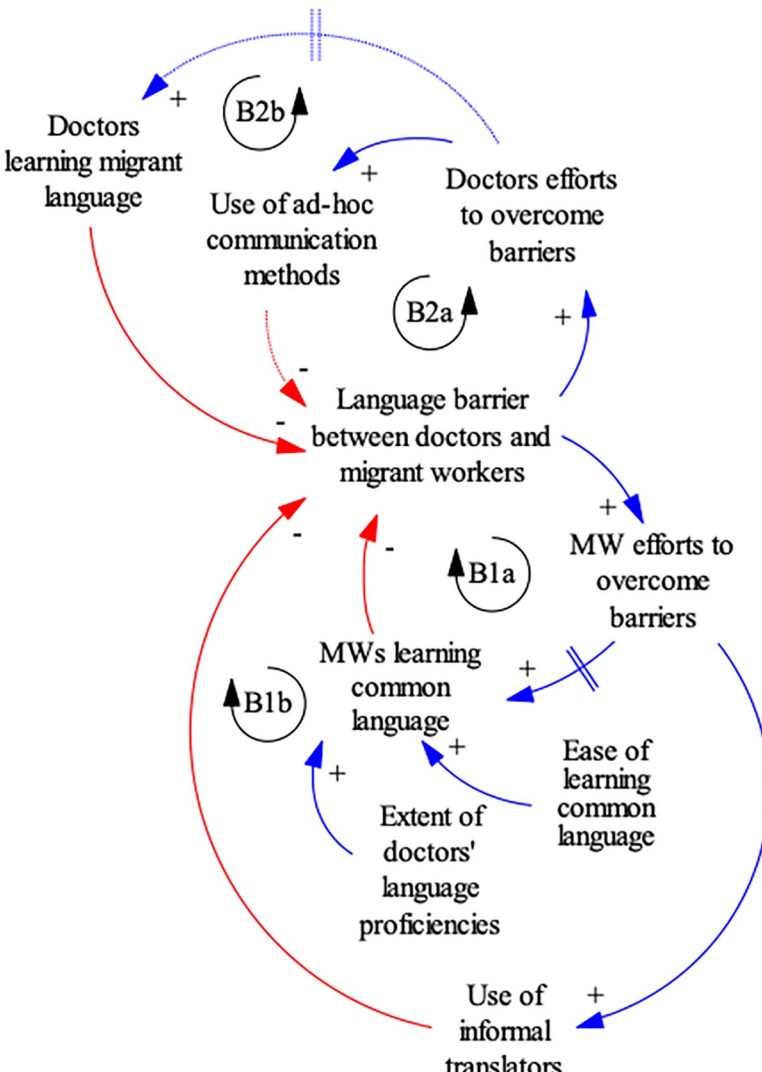

**Fig 1. Interventions to mitigate language barriers between migrant patients and health workers in Malaysia.**
B = balancing loop.

system. In the causal loop diagrams below, we also use double-barred arrows to represent delays in a response and dotted arrows to show weak linkages. Positive relationships between variables are depicted via blue arrows, and negative relationships in red arrows. The availability of one pathway may create the perception that alternative pathways are unnecessary, as seen in perceptions that it is the responsibility of migrants to learn the local language whereas the health system should allocate limited resources to the majority. The fact that many migrants do became conversant in Malay (B1a) and that migrant workers and their employers are often able to provide informal translators (B1b) reinforces this perspective. However, this overlooks the differences between different migrant groups and the varying barriers they face in overcoming linguistic and cultural barriers. Migrants becoming conversant in Malay and presence of informal interpreters, also neglects the delays in learning a common language (B1a), which creates a period of vulnerability for migrants. Similarly, if doctors think their ad-hoc communication methods of gestures and Google translate is sufficient (B2a), they may be less motivated to learn migrant languages (B2b). Based on our findings, doctors are unlikely to

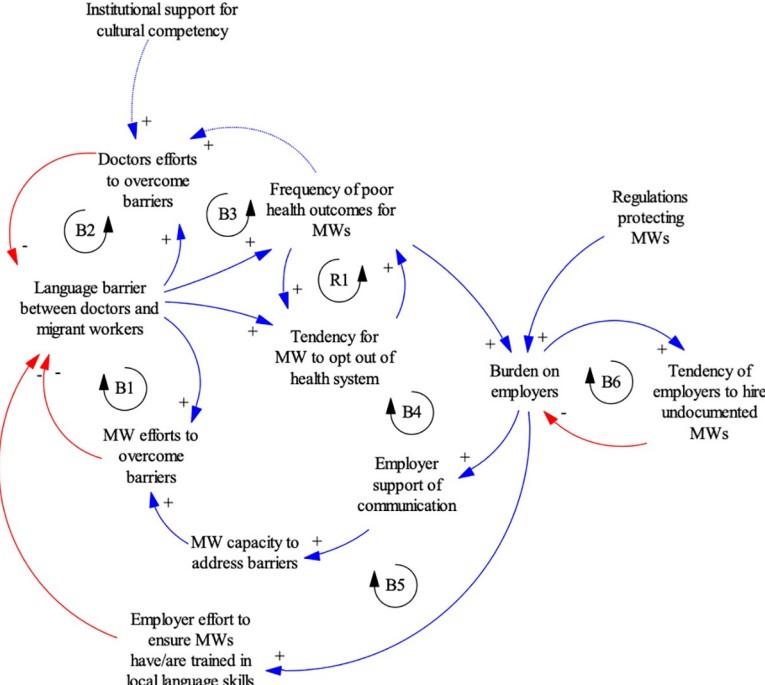

**Fig 2. Systems interactions between health workers, migrant workers and employers in Malaysia.** B = balancing loop. R = reinforcing loop.

learn migrant languages, and even when they attempt to, there is a delay given the length of time required to acquire sufficient proficiency for effective clinical communication. These ad-hoc communication methods are typically all that doctors have the capacity for and are sensible adaptive behaviours at the individual level; however, reliance on these methods masks the need for systemic solutions.

Fig 2 shows a larger system surrounding the language barrier in medical care and includes a third group, employers. Employers may not have an incentive to support migrants with language interventions, as they may fear that fluent migrants will leave to seek better paid or more skilled work than they are currently undertaking.

Migrant workers have the most incentive to overcome the language barriers (B1), but also have the most limited agency among the three groups. While some are able to, many others are not. Some may even choose to opt out of the health system in response to real or perceived problems in medical care and health outcomes (R1), to the detriment of their health and increasing the likelihood of complications in treatment when they do eventually enter the health system. This, together with the language barriers, further undermines migrant trust in the health system.

Doctors have little institutional support for cultural competency and many competing responsibilities, making B2 a largely unresponsive feedback loop. Indeed, the ad-hoc communication methods described above are typically the limit of doctors' adaptive measures in the absence of strong system incentives and supports for learning migrant languages. There is a lack of health system efforts to document evidence on language competencies and health outcomes; in the absence of data and feedback to doctors, existing poor health outcomes for migrant workers are unlikely to change doctors' efforts to acquire language competency (B3).

Employers will respond to the language barrier when migrant health outcomes pose a financial risk, by facilitating doctor-patient communication (B4) and ensuring migrants have

local language skills (B5). Increasing regulations on employers in order to protect migrants could strengthen this incentive. However, it could also have unintended outcomes if employers perceive the risks to be too high (B6): turning to undocumented workers who have few protections, or limiting hiring from countries where the language barriers are especially high (e.g. Myanmar)—which may be a desirable or undesirable outcome.

## Discussion

This paper describes how health workers, migrants and employers overcome language and cultural barriers in a health system. Use of informal interpreters was commonly reported in both countries. As per systematic review findings on care provision for migrants, informal interpreters (e.g. friends, family) are considered appropriate by providers for uncomplicated clinical situations (e.g. coughs, fever) but for mental health or serious conditions, trained interpreters are important [5]. Doctors and migrants appear to be coping without use of interpreters across primary to secondary care, but as findings indicate, serious errors and misunderstanding of procedures can occur as a result.

Lack of homogeneity in language and culture between migrants and citizens may affect health worker and employer attitudes towards migrants. In Thailand, most migrants come from neighbouring countries, with similar cultural practices and shared language in the case of Laos. Conversely, Malaysia receives migrants and refugees from further afield, with fewer cultural and linguistic similarities (except for Indonesian workers). Here, while the health system should make an attempt to provide interpreters at public hospitals, this is unlikely to be feasible for outpatient care, especially stand-alone GP clinics. Participants in our study provided practical examples of how to engage community health workers and patient navigators. Examples can be taken from industry, where the use of a community leader, with seniority and language proficiency, is common practice. As per the Thai case, the MOH in Malaysia could provide guidelines or support for training of interpreters, designed with CSOs who currently implement their own informal systems.

Structural policy constraints compound communication problems between migrants and doctors, notably the limitations of FOMEMA screening (or lack thereof) for mental health in Malaysia. Low payments for doctors for each screening, and large number of conditions they must screen for, acts as an incentive to conduct screening quickly–use of an interpreter would take time and prolong exams, so doctors use rules of thumb (visual assessments) to make diagnoses. Our study raises questions about panel doctor's competencies and training to conduct proper mental health diagnoses.

Where guidelines on cultural competence do not exist for domestically diverse populations such as in Malaysia, it is unlikely that they do for migrants. Addressing the needs of a linguistically and ethnically diverse domestic population is already a challenge in the Malaysian health system, where for example, spiritual concerns and family pressure to seek traditional therapies differ by ethnicity [47–50]. Equally, given previous tensions in Malaysia among the ethnically diverse domestic population, addressing prejudice against migrants and refugees will be challenging (82% of Malaysians wanted to see immigration levels decrease, compared to 49% in Thailand [51]). Interventions that aim to improve cultural competency among health workers, should address both migrant and domestic population groups. Here, interpreters for domestic Indian Malaysian and Chinese Malaysian populations who cannot speak Malay, could also facilitate doctor-patient communication for migrant workers from India and China. None of our health worker stakeholders had formally received cultural diversity training in the workplace or in medical school. More attention paid in curriculums and continuing professional development may help providers provide better quality of care to migrant patients.

As alluded to by a policy stakeholder and reported in other settings, making systems non-migrant friendly, by not ensuring cultural competency coupled with high user charges, may be considered a deliberate strategy of deterrence to use health services [52]. The perception that migrants will overuse services if the health system becomes more migrant-friendly is real, yet there is a paucity of evidence from LMICs that overuse occurs. Medium to low quality evidence from high-income settings suggests that while restrictive welfare and documentation policies reduce adult migrant service use, it does not for children [53,54]. In other high-income settings, migrant children's use of services was less that of native populations, except for emergency care and hospitalisation where they had higher use [55], while migrants had higher service use of emergency services and hospitalisations than non-migrants [56]. Screening and outpatient services were used less often by migrants compared to natives [56,57]. Overall, migrants tend to delay seeking healthcare until conditions are acute and costlier to treat; which raises the question of whether migrants can afford to pay high user charges even when applied [58]. Evidence from Thailand suggests that even with dedicated migrant health insurance provided by the MOPH, migrants significantly underuse both inpatient and outpatient services compared to Thai citizens, with disease status being a stronger predictor of health seeking behaviour compared to insurance status [21]. The extent to which making services more culturally competent does increase migrant service use, is a question for future research from LMICs including Thailand and Malaysia.

This study has some limitations. Participants were heterogenous with small sub-samples for policymakers, trade unions, industry and academics. We may have incurred selection bias from purposive sampling of known participants in both locations. Subsequent snowball sampling to participants beyond the researchers' known contacts, aimed to mitigate against potential selection bias. As the research topic could be considered sensitive, participants may have given socially acceptable responses. We tried to lessen potential bias with open-ended questions and inclusion of indirect questions about a third party's perspective on an issue, so participants would not feel pressured to frame responses as their own opinions. However, as the focus was on understanding cultural competency in the health system, we prioritized interviewing health professionals and those with frontline experience of treating and assisting migrants, including NGOs in Malaysia migrant health workers and volunteers in Thailand. While our qualitative findings cannot be generalized to other settings, they provide case studies assessing cultural competency of health systems for migrant service use in two LMICs. The methodological approach, of sequential qualitative work followed by systems thinking diagram conceptualization, may be useful to others considering the impact of policies in other sectors on migrant health.

## Conclusion

Migrant health policies require intersectoral thinking, even when we consider the implications of improved cultural competency and reduction of language barriers in Malaysia, there are potentially negative feedback loops for migrants. Our companion paper offers a macro-level health systems perspective on cultural competency, using the Thai case of ongoing formalisation of an existing interpreter and migrant friendly health system. Applying systems thinking in migrant health can help to identify adverse consequences of well-intentioned policies. While fears of overuse by migrants and refugees in a health system acts as a barrier against system or institutional level improvements for cultural competency, we have no existing evidence for Malaysia, Thailand, and other LMICs that this occurs. Given current system constraints, language interventions with migrants appear to be the most feasible intervention point in Malaysia, but this raises questions about who should provide or pay for language interventions (employers, the state, or migrants themselves).

## Supporting information

**S1 File. Transcription guide.**
(DOCX)

**S2 File. Topic guides.**
(DOCX)

## Acknowledgments

We thank Ms. Weixi Jiang and Dr. Shenglan Tang at Duke-Kunshan University for their administrative and technical support, and Mr. Mohd Shafiee Bidin, Ms. Nor Azlinda Abd Aziz and Ms. Nadiah Adnan at UNU-IIGH for finance and administrative support, and the IHPP administrative team for workshop organization support. We are grateful to Dr. José Siri for initial comments on the research design.

## Author Contributions

**Conceptualization:** Nicola Suyin Pocock, Tharani Loganathan, Rapeepong Suphanchaimat, Pascale Allotey, Wei-Kay Chan, David Tan.

**Data curation:** Zhie Chan, Hathairat Kosiyaporn, Wei-Kay Chan.

**Formal analysis:** Nicola Suyin Pocock, Zhie Chan, Tharani Loganathan, Rapeepong Suphanchaimat, Hathairat Kosiyaporn, Wei-Kay Chan, David Tan.

**Funding acquisition:** Nicola Suyin Pocock, Tharani Loganathan, Rapeepong Suphanchaimat, Pascale Allotey.

**Investigation:** Nicola Suyin Pocock, Tharani Loganathan, Rapeepong Suphanchaimat, Hathairat Kosiyaporn.

**Methodology:** Nicola Suyin Pocock, Zhie Chan, Tharani Loganathan, Rapeepong Suphanchaimat, Hathairat Kosiyaporn, David Tan.

**Project administration:** Nicola Suyin Pocock, Zhie Chan.

**Software:** David Tan.

**Supervision:** Nicola Suyin Pocock, Tharani Loganathan, Rapeepong Suphanchaimat, Pascale Allotey.

**Visualization:** David Tan.

**Writing – original draft:** Nicola Suyin Pocock, Tharani Loganathan, Rapeepong Suphanchaimat, Hathairat Kosiyaporn, David Tan.

**Writing – review & editing:** Nicola Suyin Pocock, Zhie Chan, Tharani Loganathan, Rapeepong Suphanchaimat, Hathairat Kosiyaporn, Pascale Allotey, Wei-Kay Chan, David Tan.

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
