## [Decision Letter · Decision Letter 0]

14 Jan 2020

PONE-D-19-28989

Moving towards culturally competent health systems for migrants? Applying systems thinking in a qualitative study in Malaysia and Thailand

PLOS ONE

Dear Dr Pocock,

Thank you for submitting your manuscript to PLOS ONE. After careful consideration, we feel that it has merit but does not fully meet PLOS ONE’s publication criteria as it currently stands. Therefore, we invite you to submit a revised version of the manuscript that addresses the points raised during the review process.

We would appreciate receiving your revised manuscript by Feb 28 2020 11:59PM. To enhance the reproducibility of your results, we recommend that if applicable you deposit your laboratory protocols in protocols.io, where a protocol can be assigned its own identifier (DOI) such that it can be cited independently in the future. For instructions see: http://journals.plos.org/plosone/s/submission-guidelines#loc-laboratory-protocols

We look forward to receiving your revised manuscript.

Kind regards,

Chaisiri Angkurawaranon

Academic Editor

PLOS ONE

Journal Requirements:

2. Please include a copy of the interview guides, in both the original language and English, as Supporting Information.

Reviewers' comments:

Reviewer's Responses to Questions

**Comments to the Author**

1. Is the manuscript technically sound, and do the data support the conclusions?

Reviewer #1: Partly

Reviewer #2: Yes

2. Has the statistical analysis been performed appropriately and rigorously? 

Reviewer #1: N/A

Reviewer #2: N/A

3. Have the authors made all data underlying the findings in their manuscript fully available?

Reviewer #1: No

Reviewer #2: Yes

4. Is the manuscript presented in an intelligible fashion and written in standard English?

Reviewer #1: Yes

Reviewer #2: Yes

5. Review Comments to the Author

Reviewer #1: The topic of your research is extremely relevant and interesting, as migrant workers' access to the health systems of host countries represent one of the key issues of concern globally. Furthermore, you focus on an geographical area (South-East Asia) which has been rarely studied on this topic; and you appropriately highlighted the importance of adding to this specific geographical focus.

Said that, I have several concerns on the current version of your manuscript:

1) The introduction. I am not convinced that starting straight from the notion of 'cultural competency' gives justice to the importance of your research topic. As a interested reader of your paper, I started the abstract and introduction with this question in mind: why should I learn what 'cultural competency is?' So, I would rather introduce the key (global and local) problem, and then which concepts and methods you use to help understanding this problem.

2) I sorely lack empirical background on who these migrant workers actually are. I believe that a rich description of the context would be necessary - especially given the qualitative nature of your work - to help the reader understand the broader dynamics taking place around the migrant workers and health professionals that you are seeking to understand. Perhaps a section before or after your methods section (and before the empirical results) would help.

3) I have many questions still unresolved on your methods. Given the sensitivity of the topics under discussion, could you please report any selection bias, strategic bias or desirability bias that you may be at risk of? Could you please reflect on your positionality as interviewer, and describe the context of the interviews themselves (e.g., when and where were they conducted, how did you seek to protect interviewees from risks of revealing sensitive information)? Also, very importantly: how did you transcribed, translated and stored the data from the interviews? Are the transcripts fully available to the readers - and if not, why? And did you engage the respondents with early or final interpretations of your findings? For example: did the respondents agree with your way of picturing the problem (and its solution) through systems dynamics? I believe that addressing these questions is critical to validate your findings.

4) Findings. While I strongly agree with your motivation for using systems dynamics to describe and address the problem (although I believe that the justification for using systems dynamics should belong to the methods section), I did not find the visualisation of your systems dynamics to capture the richness of your interview data. I would suggest to divide more clearly a) the description of the various sides of the problem from b) a proposition of the various strategies in use to address the multiple causes (or potential solutions) to the problem. An effective way to describe this approach through the use of systems dynamics may be offered from the following article:

Herrera, D., & Bleijenbergh, I. L. (2016). Cutting the Loops of Depression: a System Dynamics Representation of the Feedback Mechanisms Involved in Depression Development.

I wish the authors best of luck with further improving this manuscript.

Reviewer #2: This paper presents the results of a study to investigate notions of culturally competent health systems for migrants in LMIC such as Malaysia and Thailand. After analysing transcripts of interviews, researchers developed causal diagrams to indicate the relationships between perceptions and actions of workers, health professionals and employers.

While the paper presents some interesting work, I believe there is a point of confusion in the analysis of the paper. The results section of the paper presents thematically analysed findings of interviews, across both Malaysia (44 interviews) and Thailand (50 interviews). This is an insightful piece, as it allows the reader to learn from the more developed Thai system, and see how it can be applied in Malaysia. However, the confusion for me came on line 725 of the document (p23) which says “Based on the thematic analysis findings primarily from Malaysia…”. I am confused why the abstract, results, and discussion would be from both Malaysia and Thailand, but the causal diagram is derived primarily from Malaysia. This disparity in analysis does not seem to be properly explained in the paper. I recommend that this alignment is clarified before this article can be accepted.

Apart from that, I have the following comments:

- Line 55. Cultural competency is not a concept specifically about health services. Consider rephrasing opening sentence

- Line 60. Rephrase “Cultural competency interventions for migrant service use specifically include…”

- Line 66. Include reference for previous studies?

- Line 73 (and later). Make sure you define all acronyms on first use in the body of the text (e.g. LMIC). Once an acronym is used, continue to refer to it (e.g. p246 MHW)

- Lines 77-81. Consider revising and tie together sentences

- Lines 86-93. Tie sentences together better

- Line 141. How many years of work experience was used as selection criteria

- Line 424: “… expressed otherwise in some cases.”. rephrase

- Line 431-432. Rephrase sentence.

- Line 432 “Having a psychiatric illness…” change to “In Malaysia, having a psychiatric illness…”

- Line 592. Do you define OSH? I can’t see the definition

- Line 634-635. Rephrase last sentence of paragraph (starting “Of CSOs which ran…”)

- I would suggest moving the first paragraph (starting line 714) into the methods section (maybe around 162). This is the method of analysis used, rather than a particular result. Some of the next paragraph (starting line 725) can also be moved up and clarified – in particular the part that is talking about alternative pathways.

- Line 723 refers to dotted arrows showing weak linkages, but these are not used in your diagrams. Also include an introduction of the +, - annotations to causal diagrams in the description of feedback loops. Introduce notation of B and R within loops

- Figure 1 and 2. Are these derived from Malaysia only? If so, add something to captions

- Figure 2. Are some of these relationships missing the double bar to show delays?

- Line 783. Suggest rephrasing end of the sentence: “… - which may be a desirable or undesirable outcome”

- Line 788-792. Is the severity of a clinical situation always known from the onset?

- Paragraph starting 794. It is particularly apparent here that Thailand has been brought back into the conversation, when it was not included in the previous analysis

- Line 827. In catering for the domestic diversity for Malaysia, would this contribute to diversity in migrant populations (eg systems that cater for domestic diversity that would also serve migrants?)

- Line 833. Please clarify what “Medium to low quality evidence” means?

- Line 837. Please clarify what “except for emergency care..” means. Equal? Or higher?

6. PLOS authors have the option to publish the peer review history of their article (what does this mean?). If published, this will include your full peer review and any attached files.

Reviewer #1: Yes: Domenico Dentoni

Reviewer #2: Yes: Hannah Thinyane

---

## [Author Response · Author response to Decision Letter 0]

9 Mar 2020

We recognise the demanding and time-consuming nature of peer review, and we are very grateful to the two reviewers for their constructive feedback on our paper, which we aim to address in this resubmission. We believe the paper has significantly improved as a result. We reply to the reviewers’ specific points and concerns below.

Reviewer #1: The topic of your research is extremely relevant and interesting, as migrant workers' access to the health systems of host countries represent one of the key issues of concern globally. Furthermore, you focus on an geographical area (South-East Asia) which has been rarely studied on this topic; and you appropriately highlighted the importance of adding to this specific geographical focus.

Said that, I have several concerns on the current version of your manuscript:

1) The introduction. I am not convinced that starting straight from the notion of 'cultural competency' gives justice to the importance of your research topic. As a interested reader of your paper, I started the abstract and introduction with this question in mind: why should I learn what 'cultural competency is?' So, I would rather introduce the key (global and local) problem, and then which concepts and methods you use to help understanding this problem.

Thank you for highlighting this, we agree. We have inserted an introductory paragraph addressing the significance of the topic – 

‘Globally there are 277 million migrants and 19 million refugees, including 164 million labour migrants of whom around 30% are working in Low- and Middle-Income Countries (LMIC) [1]. LMICs Malaysia and Thailand are major destination countries for low-skilled migrants working in construction, agriculture, manufacturing, services and domestic work. Malaysia hosts an estimated 5.5 million documented and undocumented migrant workers from Indonesia, Bangladesh, Nepal and Myanmar, while Thailand hosts around 3.9 million migrants, mainly from neighbouring countries Cambodia, Laos and Myanmar [2,3]. Both countries host significant refugee populations (179,000 in Malaysia, 103,000 in Thailand) mainly from Myanmar [2,4]. Despite significant in-migration, little is known about how healthcare providers are responding to challenges posed by these changing patient demographics [5].’

2) I sorely lack empirical background on who these migrant workers actually are. I believe that a rich description of the context would be necessary - especially given the qualitative nature of your work - to help the reader understand the broader dynamics taking place around the migrant workers and health professionals that you are seeking to understand. Perhaps a section before or after your methods section (and before the empirical results) would help.

Another good point. We have included some contextual information in the introductory paragraph about where migrants come from and their sectors of work. We have also inserted contextual information about migrant healthcare financing and other barriers to access, throughout the Background section.

3) I have many questions still unresolved on your methods. Given the sensitivity of the topics under discussion, could you please report any selection bias, strategic bias or desirability bias that you may be at risk of? Could you please reflect on your positionality as interviewer, and describe the context of the interviews themselves (e.g., when and where were they conducted, how did you seek to protect interviewees from risks of revealing sensitive information)? Also, very importantly: how did you transcribed, translated and stored the data from the interviews? Are the transcripts fully available to the readers - and if not, why? And did you engage the respondents with early or final interpretations of your findings? For example: did the respondents agree with your way of picturing the problem (and its solution) through systems dynamics? I believe that addressing these questions is critical to validate your findings.

Thank you for prompting us to further elaborate details of how data were collected and the potential limitations, which were missing in the original submission. We now discuss potential selection and social desirability bias in the Discussion under Limitations –

‘We may have incurred selection bias from purposive sampling of known participants in both locations. Subsequent snowball sampling to participants beyond the researchers’ known contacts, aimed to mitigate against potential selection bias. As the research topic could be considered sensitive, participants may have given socially acceptable responses. We tried to lessen potential bias with open-ended questions and inclusion of indirect questions about a third party’s perspective on an issue, so participants would not feel pressured to frame responses as their own opinions.’

We have provided further detail on interviewer positionality, transcription and storage in Methods > Data collection & sampling –

‘Interviews were primarily conducted by a team of medical doctors and academic researchers in both countries. Interviewers could be perceived as trusted authority figures, particularly with migrant workers, MHW and MHV. To lessen potential power imbalances between researchers and participants, the majority of interviews were conducted in locations and at times of the participants’ choosing, in a space they were comfortable in. We emphasized that anonymity and confidentiality would be maintained in study reporting. Migrant participants especially were assured that they could refuse to answer questions or to end the interview at any time. 

Interviews were transcribed verbatim and analysed in native languages by the multi-lingual research team. Audio files and electronic transcripts were stored on secure servers, and transcripts were stored securely in locked cupboards in the researcher’s offices. In Malaysia, all except one interview were conducted in English, while interviews with MHW and MHV were conducted in Thai. Following analysis in both countries, selected quotes were translated to English for presentation in this manuscript and accompanying papers. Given the perceived sensitive nature of the research and to encourage participation, participants were not asked for consent to have their transcripts available beyond the immediate research team.’ 

We did not conduct formal validation of systems thinking diagrams with participants. However, findings were based on triangulated interview findings and underwent several iterations, discussed extensively within the research team. We presented the final versions in a workshop in Malaysia, now mentioned in Methods > Systems thinking diagrams – 

‘The systems diagrams presented in this paper draw on Malaysian data, which were shared with interview participants at a dissemination workshop in Kuala Lumpur in December 2018 [43].’

4) Findings. While I strongly agree with your motivation for using systems dynamics to describe and address the problem (although I believe that the justification for using systems dynamics should belong to the methods section), I did not find the visualisation of your systems dynamics to capture the richness of your interview data. I would suggest to divide more clearly a) the description of the various sides of the problem from b) a proposition of the various strategies in use to address the multiple causes (or potential solutions) to the problem. An effective way to describe this approach through the use of systems dynamics may be offered from the following article:

Herrera, D., & Bleijenbergh, I. L. (2016). Cutting the Loops of Depression: a System Dynamics Representation of the Feedback Mechanisms Involved in Depression Development.

Thank you for these helpful suggestions and the reference. We have shifted justification for using systems thinking diagrams to the Methods section under its own heading. As mentioned above, the final versions had undergone several iterations based on the themes in the data. We understand the point about distinction between the problem identification vs. solutions vs. causes, which we believe are broadly represented in the qualitative sub-themes. However, we are not sure it is necessary to breakdown the systems diagrams into these components. We aimed to distil the qualitative themes into simplified, aggregate variables for clarity in presentation in the diagrams, with instead rich detailed presentation of qualitative findings in that section. We believe that the current diagrams satisfy the aims of our analysis, and are happy to further justify this in the text if the reviewer believes this is necessary.

I wish the authors best of luck with further improving this manuscript.

Thank you, and thanks again for taking the time to review our paper.

Reviewer #2: This paper presents the results of a study to investigate notions of culturally competent health systems for migrants in LMIC such as Malaysia and Thailand. After analysing transcripts of interviews, researchers developed causal diagrams to indicate the relationships between perceptions and actions of workers, health professionals and employers.

While the paper presents some interesting work, I believe there is a point of confusion in the analysis of the paper. The results section of the paper presents thematically analysed findings of interviews, across both Malaysia (44 interviews) and Thailand (50 interviews). This is an insightful piece, as it allows the reader to learn from the more developed Thai system, and see how it can be applied in Malaysia. However, the confusion for me came on line 725 of the document (p23) which says “Based on the thematic analysis findings primarily from Malaysia…”. I am confused why the abstract, results, and discussion would be from both Malaysia and Thailand, but the causal diagram is derived primarily from Malaysia. This disparity in analysis does not seem to be properly explained in the paper. I recommend that this alignment is clarified before this article can be accepted.

Thank you for highlighting this. We struggled with the presentation of our findings. During analysis, we realised that it would not be possible to reconcile both country contexts into a unified systems thinking diagram, which rely on rich (local) contextual information to accurately represent phenomena. Instead, we opted to write two papers – this paper focussing on Malaysian findings with corresponding systems diagrams, and a second companion paper (submitted elsewhere) focussing on Thai findings with a corresponding systems diagram. We have now clarified this, in Methods > Systems thinking diagrams - 

‘In this paper, we present qualitative findings for Malaysia and Thailand as part of a comparative study of two countries at very different stages of migrant-friendly health systems development. Because of this difference, and given the local contextual complexity, we decided to elaborate systems thinking diagrams for Malaysia only in this paper. The Malaysian health system has no formal cultural competency provisions currently, compared to Thailand, where a semi-formalized interpreter system is in place.

The systems diagrams presented in this paper draw on Malaysian data, which were shared with interview participants at a dissemination workshop in Kuala Lumpur in December 2018 [43]. A companion paper submitted elsewhere offers a macro-level health systems perspective on cultural competency, using the Thai case of ongoing formalisation of an existing interpreter and migrant friendly health system [45].’

Apart from that, I have the following comments:

Thank you for the detailed comments below. We have made revisions.

- Line 55. Cultural competency is not a concept specifically about health services. Consider rephrasing opening sentence

Thank you for the point. We agree and have introduced a clarifying sentence – 

‘Cultural competence is a concept that acknowledges the importance of culture as fundamental to effective communication and interaction within a multicultural environment. In healthcare, cultural competency describes interventions that aim to improve accessibility and effectiveness of health services for people from ethnic minority backgrounds [6].’

- Line 60. Rephrase “Cultural competency interventions for migrant service use specifically include…”

We think this sentence is clear as is.

- Line 66. Include reference for previous studies?

Reference 3 has been added.

- Line 73 (and later). Make sure you define all acronyms on first use in the body of the text (e.g. LMIC). 

Once an acronym is used, continue to refer to it (e.g. p246 MHW)

Thank you, we have corrected the acronyms throughout.

- Lines 77-81. Consider revising and tie together sentences

We think these sentences are clear as is.

- Lines 86-93. Tie sentences together better

We prefer to keep sentences short for clarity.

- Line 141. How many years of work experience was used as selection criteria

We have clarified in the text. Selection criteria included those who worked less than 2 years, and those who worked for more than 2 years, based on the average of turnover rate of 2 years. 

- Line 424: “… expressed otherwise in some cases.”. rephrase

Rephrased to ‘Migrant CSOs or TU’s disagreed in some cases.’

- Line 431-432. Rephrase sentence.

Rephrased to ‘Screening for psychiatric illnesses is conducted as part of the mandatory medical screening process for incoming documented migrant workers, which was implemented by FOMEMA, a company appointed by the government to conduct foreign worker’s medical screening.’

- Line 432 “Having a psychiatric illness…” change to “In Malaysia, having a psychiatric illness…”

Thank you for the suggestion – we have revised.

- Line 592. Do you define OSH? I can’t see the definition

Revised to include the definition where OSH first appears.

- Line 634-635. Rephrase last sentence of paragraph (starting “Of CSOs which ran…”)

Rephrased to ‘In CSO clinics with informal interpreter systems, continuing training consisted of talks by specialists.’

- I would suggest moving the first paragraph (starting line 714) into the methods section (maybe around 162). This is the method of analysis used, rather than a particular result. Some of the next paragraph (starting line 725) can also be moved up and clarified – in particular the part that is talking about alternative pathways.

Thank you for the suggestion, we have moved the systems thinking description to the Methods section. We think it best to keep the part about alternative pathways here, so readers (unfamiliar with systems thinking) understand the next few sentences describing the findings.

- Line 723 refers to dotted arrows showing weak linkages, but these are not used in your diagrams. Also include an introduction of the +, - annotations to causal diagrams in the description of feedback loops. Introduce notation of B and R within loops

We have included a key under each figure for B and R notation. We have included a sentence about the positive and negative representation in feedback loops here (we prefer to keep this separate to the subsequent descriptions for each feedback loop, which have mixed + and – relationships within them). The dotted arrows are there, but faint (this appears to be a limitation of the software).

- Figure 1 and 2. Are these derived from Malaysia only? If so, add something to captions

Thank you, we have added Malaysia to the captions.

- Figure 2. Are some of these relationships missing the double bar to show delays?

No, we do not have delays to depict in Figure 2.

- Line 783. Suggest rephrasing end of the sentence: “… - which may be a desirable or undesirable outcome”

We believe this is clear as is.

- Line 788-792. Is the severity of a clinical situation always known from the onset?

No, but we do not think it adds to the discussion to go into this caveat. The review findings cited offer an indication of what providers think is appropriate re. informal vs. formal interpreters, which we think is sufficient.

- Paragraph starting 794. It is particularly apparent here that Thailand has been brought back into the conversation, when it was not included in the previous analysis

Thank you for highlighting this. While Thailand is not formally represented in the diagrams, they are included in the prior qualitative findings. As this is the Discussion, we believe it is appropriate to reflect on the situation in other countries including Thailand, as one of our study countries.

- Line 827. In catering for the domestic diversity for Malaysia, would this contribute to diversity in migrant populations (eg systems that cater for domestic diversity that would also serve migrants?)

Thank you for this insightful point. The answer is partially yes, for Indian and Chinese migrant workers (both ethnic groups are also Malaysians but there are cultural differences remaining, except perhaps language). We have included a sentence about this here – 

‘Here, interpreters for domestic Indian Malaysian and Chinese Malaysian populations who cannot speak Malay, could also facilitate doctor-patient communication for migrant workers from India and China.’

- Line 833. Please clarify what “Medium to low quality evidence” means?

The authors used risk of bias guidelines from the Grading of Recommendations Assessment, Development and Evaluation (GRADE) approach – evidence ratings are commonly used in public health. We have added a citation to GRADE guidelines here. We think it best not to go into detail in the text about the tool used.

- Line 837. Please clarify what “except for emergency care..” means. Equal? Or higher?

Thank you for the question – we have clarified ‘where they had higher use’.

---

## [Editor Report · Decision Letter 1]

18 Mar 2020

Moving towards culturally competent health systems for migrants? Applying systems thinking in a qualitative study in Malaysia and Thailand

PONE-D-19-28989R1

Dear Dr. Pocock,

We are pleased to inform you that your manuscript has been judged scientifically suitable for publication and will be formally accepted for publication once it complies with all outstanding technical requirements.

With kind regards,

Chaisiri Angkurawaranon

Academic Editor

PLOS ONE
---

## [Editor Report · Acceptance letter]

23 Mar 2020

PONE-D-19-28989R1 

Moving towards culturally competent health systems for migrants? Applying systems thinking in a qualitative study in Malaysia and Thailand 

Dear Dr. Pocock:

I am pleased to inform you that your manuscript has been deemed suitable for publication in PLOS ONE. Congratulations! Your manuscript is now with our production department. 

With kind regards,

on behalf of

Dr. Chaisiri Angkurawaranon 

Academic Editor

PLOS ONE